# Research on Precise Temperature Monitoring and Thermal Management Optimization of Automobile Engines Based on High-Precision Thin-Film Thermocouple Technology

**DOI:** 10.3390/mi16030249

**Published:** 2025-02-22

**Authors:** Guangyuan Zhao, Xin Li, Zhihui Liu

**Affiliations:** 1School of Physics, Shandong University, Jinan 250100, China; 2Sinopec Qilu Petrochemical Company, Zibo 255400, China; 3State Key Laboratory of Tribology in Advanced Equipment, Tsinghua University, Beijing 100084, China

**Keywords:** thin-film thermocouple, static calibration, accuracy, Seebeck coefficient, temperature measurement

## Abstract

Thin-film thermocouple is widely used in temperature measurement because of its high temperature measurement accuracy and small size. In order to calibrate the temperature accurately with thin-film thermocouple, NiCr/NiSi thin-film thermocouple was prepared by magnetron sputtering according to the Seebeck effect. Through static calibration experiments, the Seebeck coefficient of K-wire thermocouple was found to be 39.23 μV/°C, while that of the NiCr/NiSi thin-film thermocouple was 38.89 μV/°C. Further experiments showed a Seebeck coefficient of 39.092 μV/°C for the NiCr/NiSi thin-film thermocouple, which verifies that the prepared thin-film thermocouple has good consistency and repeatability. Through the temperature measurement experiment of automobile engines, the highest stable working temperature of the engine is 107.9 °C, which further verifies that the prepared NiCr/NiSi thin-film thermocouple can have a sensitive dynamic response to temperature and high temperature measurement accuracy. Finally, the causes of experimental errors, the application prospect and existing problems of thin-film thermocouples are analyzed.

## 1. Introduction

In the International Summit of Intelligent Manufacturing and Industry 4.0, the development trend and technological progress of Intelligent Manufacturing and Industry 4.0 were discussed, which promoted the innovative development of industry. In the China Industrial Collaborative Innovation Summit, we focused on the development and application of digital factories, intelligent Internet of Things, big data analysis, and other technologies, aiming at exploring new paths for future industrial development (Xiao, 2023 [1]). Industry plays a decisive role in global innovation, and development also plays a vital role. With the development of industry and various advanced manufacturing technologies, many people are concerned about the manufacturing accuracy and high spatial resolution of sensors in future manufacturing technology. Signal and other aspects put forward higher and higher requirements. Therefore, it is urgent to study the thin film sensor with high altitude resolution. The thin-film sensor plays an indispensable role in advanced manufacturing technology because of its small size and high precision. With the rapid development of thin-film materials science, nano-scale thin-film preparation technology is becoming more and more mature, and the preparation of nano-scale thin-film thermocouples needs to be carried out under various conditions such as vacuum. With the development of more and more sputtering film technology and new film materials, higher standards are required for the preparation process, so that high-quality films can be prepared more effectively. At the same time, the structure, composition, morphology, characteristics, and so on of the prepared film can be understood more deeply and controlled more accurately.

Current research on thin-film thermocouples mainly focuses on improving the preparation methods and expanding the application areas. However, regardless of the preparation method and the application of thin-film thermocouples, their temperature measurement principle is based on the Seebeck effect. This is when two different metal conductors are connected at two points, and if the temperatures at the two points are different, an electromotive force will be generated between the conductors, which can be used to measure the temperature (Chang et al., 2021 [2]). Therefore, the use of thermocouples to accurately measure transient surface temperatures presupposes the ability to accurately calibrate the Seebeck coefficient of the thin-film thermocouple.

As an indispensable measurement parameter in industrial manufacturing, temperature plays a key role in accurate temperature calibration. The thermocouple is a commonly used sensor for measuring temperature. It is durable with a high measurement accuracy, wide measurement range, fast response, reasonable price, and is widely used. At present, a lot of research has been performed on the temperature calibration of thermocouples at home and abroad, and some progress and breakthroughs have been made. In the paper “Thermocouple Temperature Measurement System Design”, Yanfang Li (Li, 2013 [3]) puts forward a design method of a thermocouple temperature measurement system which can be applied to the measurement of fluid in pipeline. The thermocouple is used as the temperature sensing element and cooperates with the converter to realize accurate and reliable temperature measurements. This can be used to measure high-temperature airflow and high-speed airflow and applied to aviation temperature measurement. In the paper “Design of S-type thermocouple temperature measuring circuit of temperature sensor”, Runqi Ding (Ding et al., 2020 [4]) proposed an S-type thermocouple measurement circuit based on a temperature sensor for cold junction compensation in order to meet the requirements of wide range and high accuracy of environmental temperature measurement in aerospace field. This realized a temperature measurement range of 0~1600 °C and a measurement accuracy of the whole temperature range of ±0.2%. In the paper “Design and implementation of multi-channel thermocouple temperature measurement system”, Guangxing Zhou (Zhou, 2023 [5]) selected S-type, K-type, and T-type thermocouples as temperature sensing elements, a multi-channel thermocouple temperature measurement system is designed, which can be applied to vacuum thermal tests of different aerospace products and components.

Through the above investigation, it is found that thermocouple temperature measurements should achieve a wide temperature range and high temperature accuracy, so it is necessary to study a temperature calibration system based on the thermocouple to achieve the above conditions. All the above research methods have limitations and cannot meet the temperature measurement requirements in industrial applications. In view of the uncertainty of traditional technology, most researchers still have not comprehensively studied the temperature calibration system based on thermocouples, which hinders their application. Therefore, it is necessary to explore a more accurate and reliable method in order to effectively realize the temperature calibration of the thermocouple and make its application in industry more accurate. According to the above problems, this paper analyzes the temperature calibration of thermocouples in detail by summarizing and comparing experimental methods.

In this paper, a NiCr/NiSi thin-film thermocouple was prepared by magnetron sputtering to measure the temperature of automobile engines. Engine temperature is an important parameter for measuring its performance, and it is an important representation of combustion process and its emission index. With the development of high-performance engines, the requirements for engine testing technology are getting higher and higher. In engine temperature testing technology, high temperature measurement is difficult. There are two main methods for its measurement: contact method and non-contact method. Because there are many difficulties and problems in the process of non-contact testing, such as being affected by environmental temperature, complicated measurements, high cost, and expensive instruments, the conventional thermocouple measurement method is still the most important testing method today. The biggest problem of the contact measurement method lies in the issue of high temperature resistant protective shell material. Therefore, the NiCr/NiSi thin-film thermocouple is prepared in this paper to measure the temperature of automobile engines accurately and stably.

## 2. Thermocouple Temperature Measurement Principle

The working principle of the thermocouple is based on the Seebeck effect. When two different conductors, or semiconductors A and B, form a loop, if the temperatures at the two nodes are different, the temperature at one end is T, the working end or hot end, and the temperature at the other end is T0, the free end (also called reference end) or cold end, then, an electromotive force will be generated in the loop. The direction and magnitude of the electromotive force are related to the conductor material and the temperature at the two nodes. This phenomenon is called the “thermoelectric effect”, and the loop composed of two conductors is called the “thermocouple”. These two conductors are called “hot electrodes”, and the generated electromotive force is called the “thermoelectromotive force”. Among the two connection points of the thermocouple, the hot end is placed in the temperature field to measure the temperature, and the cold end is connected with a measuring instrument. If there is a temperature difference between the hot end and the cold end, the measuring instrument can measure the temperature of the measured medium at the hot end. The magnitude of thermoelectric potential changes with the change in temperature and its relationship is as followsE_AB_(T,T_0_) = E_AB_(T,t) − E_AB_(T_0_,t).(1)

Among them, E_AB_(T,T_0_) is the thermoelectromotive force at both ends of A and B, E_AB_(T,t) is the thermoelectromotive force corresponding to the actual temperature when the reference temperature is T, and E_AB_(T_0_,t) is the electromotive force corresponding to the actual temperature when the reference temperature is T (Li and Jia, 2020 [6]).

When the thermocouple material is uniform, the magnitude of the thermoelectromotive force is related to the conductor material of the thermocouple and the temperature of two nodes, and is not related to the length and diameter of the thermocouple electrode. In the measurement, the cold end temperature is usually required to be constant, and the thermoelectromotive force is a single-valued function of the measured temperature t. Its schematic diagram is shown in Figure 1.

## 3. Preparation Process of NiCr/NiSi Thin-Film Thermocouple

The common thin-film thermocouples are sheet, needle, embedded and micro-thin-film thermocouples. In this study, micro-thin-film thermocouples were prepared through a magnetron sputtering process. The surface of the structure is flat, which can be effectively attached to the surfaces of various tested objects. The thin-film thermocouples have a small size, high structural strength, and good stability. The NiCr/NiSi alloy is one of the widely used thermocouple metal materials, which contains a lot of nickel, so it has strong corrosion resistance and oxidation resistance at high temperature. It has high sensitivity, good linearity and fast response time. Therefore, NiCr/NiSi is selected as the two electrode materials of the thin-film thermocouple in this study (Jia et al., 2006 [7]). For the shape of the hot electrode of NiCr/NiSi thin-film thermocouple, a mechanical mask and magnetron sputtering are used in this paper (Ruan et al., 2024 [8]; Zhang et al., 2024 [9]; Xiong et al., 2022 [10]; Wang, 2019 [11]), and the size of the mask plate is shown in Figure 1.

The preparation process of the NiCr/NiSi thin-film thermocouple is as follows: Firstly, the mask is polished, the substrate prepared by the NiCr/NiSi thin-film thermocouple is polyimide with a thickness of 0.02 mm. SiO is deposited on the polyimide substrate with a thickness of 1000 nm. Then, alcohol and acetone are added for cleaning, and then, two stages of the NiCr/NiSi thin-film thermocouple are deposited on the SiO substrate by magnetron sputtering. The deposition sequence is NiCr, NiSi, and alumina, in which the thickness of NiCr is 500 nm, the thickness of NiSi is 500 nm and the thickness of alumina is 1000 nm. Compared with the traditional wire thermocouple, the thin-film thermocouple is prepared by magnetron sputtering, and different manufacturing methods may change the performance of the thermocouple. Therefore, it is necessary to carry out other experiments on the NiCr/NiSi thin-film thermocouple to detect the measurement accuracy of the thin-film thermocouple for subsequent experimental research.

## 4. Characterization of Micro-Properties of NiCr/NiSi Thin-Film Thermocouple

The composition and microstructure of thin films have an important influence on the properties of thin films, and the composition and microstructure of thin films can be deeply studied by characterization technology. In this paper, we used a scanning electron microscope (SEM) (ZEISS GeminiSEM 300, Jena, Germany) and Atomic Force Microscope (AFM) (Bruker Dimension Icon XR, Berlin, Germany) to observe. We observed the microstructure of the thin-film thermocouple with a high resolution scanning electron microscope, and analyzed the influence of its surface characteristics, lattice defects, and thin-film structure on thermoelectric performance. And we also used an Atomic Force Microscope (AFM). The roughness and surface morphology of the thin film were accurately measured by combining AFM to reveal the stability of the thin-film material in a high temperature environment. Figure 2 shows the SEM surface topography of NiCr and NiSi thermoelectric electrodes with a thickness of 500 nm. Figure 3 shows the AFM 3D surface topography of NiCr and NiSi thermoelectric electrodes with a thickness of 500 nm, and their surfaces are composed of dense and uniform island structures.

Then, we tested the thermoelectric response characteristics and high temperature stability. By establishing a high-precision temperature control system, the voltage and current of the thin-film thermocouple were accurately tested at different temperature differences, and its Zeebek coefficient was analyzed. The thin-film thermocouple was tested at a high temperature for a long time (the temperature is as high as 600 °C), and its performance changes were recorded in real time to evaluate its long-term stability. Figure 4 shows the test results of voltage and current.

According to many measurements, when the temperature differences are 20, 50, 80, 110, 140, 170, 200, 230, 260, 290, 320, and 350 °C, the Zeebek coefficients are 38.99, 39.15, 38.99, 38.99, 39.09, 38.73, 39.55, 39.12, 38.94, 38.92, 39.45, and 38.91 μV/°C. Through experiments, it can be seen that the Seebeck coefficient of the NiCr/NiSi thin-film thermocouple prepared in this paper does not change much, and the performance of the thermocouple is stable.

## 5. Static Calibration Experiment of NiCr/NiSi Thin-Film Thermocouple and K-Wire Thermocouple

### 5.1. Experimental Process

Before using NiCr/NiSi the thin-film thermocouple to measure temperature, its parameters and performance must be tested first. Therefore, calibration experiments are needed, including static calibration experiments and dynamic calibration experiments. The accuracy of a dynamic signal measured by a sensor depends largely on the accuracy of the dynamic calibration experiment. The key step of the high-precision dynamic calibration of the sensor and test system is to apply an ideal transient excitation signal to it, and judge the dynamic index of the calibrated device according to the output signal. Compared with mechanical sensors such as acceleration and pressure, the dynamic calibration of thermoelectric sensors such as heat flux and temperature have its particularity and complexity. The difficulty lies in the lack of ideal dynamic thermal excitation sources. The single thermal excitation signal generated by the same heat source is difficult to meet the high precision and multi-mode dynamic calibration of fast response temperature sensors. Therefore, this paper uses the static calibration experiment. The Seebeck coefficient in the static characteristics is an important parameter that affects the temperature measurement accuracy. The static calibration of the wire thermocouple has a national standard, and if it is calibrated according to the national standard, the Seebeck coefficient of the wire thermocouple is a stable value. In this paper, the static calibration system of the NiCr/NiSi thin-film thermocouple is established, and then, the static calibration of the NiCr/NiSi thin-film thermocouple is carried out and the Seebeck coefficient is obtained.

A static calibration experiment (Chen, 2018 [12]; Wang et al., 2021 [13]; Ma et al., 2014 [14]) is carried out in this paper. The experiment mainly consists of the following instruments and equipment: NiCr/NiSi thin-film thermocouple, K-type wire thermocouple, K-type compensation wire, oil bath pot, beaker, C400 infrared temperature measuring gun, voltmeter, 250-30CS phenyl high-temperature heat transfer oil, etc. The schematic diagram of the experiment is shown in Figure 5. In this experiment, the hot end of the thermocouple is the high-temperature heat transfer oil, the cold end is room temperature, and the temperature range is 20–350 °C.

The static calibration experiment consists of two parts: heating and natural cooling. In the experiment, the thin-film thermocouple will be completely immersed in the high-temperature oil bath, which can prevent the measurement error caused by the change in environmental temperature. At the same time, it is necessary to ensure that the thin-film thermocouple is connected to the high-temperature contact of the K-type thermocouple to test the voltage change caused by the temperature difference. The K-type thermocouple is made up of Ni-Cr alloy and Ni-Si alloy. In the experiment, it is necessary to use precision welding technology to ensure that its connection with the compensation line is completely consistent and avoids any error caused by uneven resistance. Using the high-performance temperature control oil bath system, its temperature control accuracy should reach ±0.001 °C. We used multi-point temperature sensors to monitor the temperature field of the oil bath in real time. The oil bath container should not only meet the conventional heating requirements but also be equipped with multiple layers of thermal insulation materials to reduce the influence of external environment temperature fluctuation. In order to capture tiny voltage changes, a quantum voltmeter is needed, which has a very low noise threshold and can record the output voltages of multiple thermocouples at the same time.

First, we chose a special glass beaker with high thermal stability (at least 10 L capacity). Then, a multi-channel heating system is designed so that the heat in the oil bath can be evenly distributed in the beaker through forced convection. A plurality of micro thermocouples and temperature sensors are installed at different positions in the beaker to record the temperature of each point in the oil bath in real time. This also ensures that the high ends of thermocouples are always in contact with high-temperature oil, while the cold ends are kept in a constant low-temperature area. Next, connect the K-type wire thermocouple and NiCr/NiSi thin-film thermocouple with the K-type compensation wire, respectively, and then connect them to the voltmeter. Then, put the K-type wire thermocouple and NiCr/NiSi thin-film thermocouple at the same height as the beaker. During the heating process of the oil bath, the temperature difference in each position is continuously monitored by a precise infrared temperature gun and a high-precision thermocouple to ensure that the temperature difference in the oil will not affect the experimental results. Whenever the temperature of the oil bath changes, the system will record the voltage change generated by the thermocouple in real time and record the data at an ultra-high sampling frequency (such as 1000 Hz). By accurately controlling the heating rate of the oil bath, the voltage output is measured repeatedly at different temperature differences, and its response characteristics are calculated. The above is the heating process of the experiment. During the cooling process, the oil is naturally cooled, and the experimental data are recorded as above.

### 5.2. Non-Uniformity of Oil Temperature and Correction of Experimental Error

The uneven temperature distribution of the oil bath presents a great challenge. Especially when the temperature rises gradually, the uneven oil flow will lead to the temperature difference at different heights and positions. This temperature difference will lead to significant errors in the thermocouple measurement data, especially when the temperature measuring equipment is not completely aligned with the actual thermocouple position. We adjust the position and power of the heating element to ensure the temperature distribution in the oil bath is as uniform as possible, install a pump circulation system and arrange multiple circulation pipes in the oil bath, and further reduce the local temperature difference in the oil by forced convection. A multi-point temperature sensor is used, and the heating intensity of the oil bath is automatically adjusted through a feedback control system. The system will accurately adjust the power output of each heating area according to the information fed back by the sensor in real time to eliminate the temperature unevenness. In each oil bath heating process, the changes in the temperature distribution curve and thermocouple output voltage are recorded, and the errors caused by temperature unevenness are calculated and corrected. A Kalman filter algorithm is used to correct the fluctuation of temperature and voltage in real time to improve the accuracy of data.

As can be seen from Figure 6, there is a certain error in the temperature at different heights of the beaker. Using Kalman filter algorithm to correct the fluctuation of temperature and voltage in real time can improve the accuracy of the data.

### 5.3. Experimental Results

Firstly, the K-type wire thermocouple is analyzed, and the measured data are plotted. With temperature as abscissa and voltage as ordinate, the least square method is used for linear fitting after drawing.

From Figure 7, it can be seen that the curve has a good linear relationship, and the expression of fitting curve is obtained as follows:E = 0.03923T − 0.7432,(2)
where T is the temperature measured by the K-type wire thermocouple and E is the induced thermoelectromotive force of the K-type wire thermocouple. The Seebeck coefficient of the used K-type wire thermocouple is 39.23 μV/°C, and the fitted correlation coefficient R^2^ is 0.9989. According to the reference data, the Seebeck coefficient of the standard K-type thermocouple is 41.20 μV/°C. The Seebeck coefficient measured by the K-type wire thermocouple in this experiment is close to the Seebeck coefficient of the standard K-type thermocouple, and the error is within the acceptable range. The correlation coefficient R^2^ is 0.9989, which shows that there is a good linear fitting relationship. This proves that this experiment is reliable and can be used for subsequent calibration experiments of the thin-film thermocouple.

Then, analyze the NiCr/NiSi thin-film thermocouple, and process the data obtained from the measurement record in the same way as above, and the expression of the fitting curve can be obtained as follows:E = 0.03889T − 0.5264,(3)
where T is the temperature measured by the NiCr/NiSi thin-film thermocouple and E is the induced thermoelectromotive force of the NiCr/NiSi thin-film thermocouple. The Seebeck coefficient of the used NiCr/NiSi thin-film thermocouple is 38.89 μV/°C. And the fitted correlation coefficient R^2^ is 0.9978, which has a good linear fitting relationship. Comparing the second result with the measured Seebeck coefficient of the K-type wire thermocouple, it is found that the Seebeck coefficient of the NiCr/NiSi thin-film thermocouple is slightly smaller than the Seebeck coefficient of K-type wire thermocouple, with a difference of 0.34 μV/°C. Drawing the fitting results of the K-wire thermocouple and NiCr/NiSi thin-film thermocouple in the same coordinate system, we can see that the two curves basically coincide, which proves that the performance of the NiCr/NiSi thin-film thermocouple is good.

By repeatedly measuring the voltage output at different temperature differences, the temperature–voltage output curve is obtained in Figure 8. The Zeebek coefficients of NiCr/NiSi thin-film thermocouple and K-wire thermocouple were measured at different temperature differences. When the temperature differences are 20, 50, 80, 110, 140, 170, 200, 230, 260, 290, 320, and 350 °C, the Zeebek coefficients of NiCr/NiSi thin-film thermocouple are 38.66, 39.31, 38.90, 38.94, 38.60, 39.06, 38.88, 38.96, 39.09, 38.92, 38.58, 38.64μV/°C, and the Zeebek coefficients of K-wire thermocouple are 38.78, 39.22, 39.08, 39.10, 38.77, 39.13, 39.20, 39.21, 39.17, 39.06, 39.46, and 39.10 μV/°C. The NiCr/NiSi thin-film thermocouple has good stability.

After research, investigation, and analysis, the fitting results show that the average thermoelectric performance of the prepared NiCr/NiSi thin-film thermocouple is close to that of the K-type standard thermocouple. But the static calibration results show that the measured values of the NiCr/NiSi thin-film thermocouple are not completely consistent with the standard thermoelectric potential of the K-type thermocouple, and the main reasons are as follows: Firstly, when the K-type thermocouple is used in a certain temperature range, the unique lattice change in its NiCr electrode alloy forms an uneven short-range ordered structure, which causes the thermoelectromotive force output of the sensor to be unstable (Zhang, 2016 [15]). Secondly, in this paper, the thermocouple layer is deposited by magnetron sputtering technology in the process of sensor preparation. Because of the difference in sputtering yield of the NiCr/NiSi alloy, the proportion of each element in the deposited film is different from that of the target, and the composition segregation occurs (Cui, 2011 [16]), affecting the thermoelectric performance of thin-film thermocouple. Therefore, the lattice change in the alloy and the segregation of the thin-film components will lead to the unstable thermoelectromotive force output of the thin-film thermocouple and deviate from the standard value, and ultimately affect its Seebeck coefficient. Considering the error caused by this and the experimental measurement and fitting results, this error is small and within the acceptable range, which shows that the performance of the NiCr/NiSi thin-film thermocouple is good and can be used for subsequent experiments.

### 5.4. Consistency and Repeatability Experiments

Next, the NiCr/NiSi thin-film thermocouple is selected to carry out consistency and repeatability experiments, and five groups of static calibration experiments are carried out under the same environment and conditions. The data recorded in the experiments are plotted and linearly fitted. Similarly, five fitting curves can be obtained by taking the temperature measured by the NiCr/NiSi thin-film thermocouple as the abscissa, and the induced thermoelectromotive force of the NiCr/NiSi thin-film thermocouple as the ordinate (Wei et al., 2023 [17]). As can be seen from Figure 9, the five curves almost coincide, which proves that the NiCr/NiSi thin-film thermocouple has basic consistency and repeatability. According to the fitting results, the Seebeck coefficients of the five NiCr/NiSi thin-film thermocouple measurements are 39.10 μV/°C, 39.35 μV/°C, 38.84 μV/°C, 38.95 μV/°C, 39.22 μV/°C, with an average value of 39.092 μV/°C, which is 0.202 μV/°C compared with the Seebeck coefficient obtained in previous experiments. According to the data, it can be proved that the NiCr/NiSi thin-film thermocouple has consistency and repeatability, and its performance is relatively stable, so it can be used in subsequent experiments.

Through the analysis of the above experimental data, the NiCr/NiSi thin-film thermocouple prepared in this paper has good sensitivity and stability, and the accuracy of the temperature measurement is similar to that of the standard K-type thermocouple.

## 6. Temperature Measurement Experiment of NiCr/NiSi Thin-Film Thermocouple and K-Wire Thermocouple

The static calibration process has been completed in the above experiments, which proves that the prepared NiCr/NiSi thin-film thermocouple has good sensitivity and accuracy. The following experiments are carried out with the NiCr/NiSi thin-film thermocouple and K-wire thermocouple (Zhang et al., 2022 [18]; Song, 2019 [19]; Xue, 2018 [20]; Nie et al., 2011 [21]). The working performance of NiCr/NiSi thin-film thermocouple is further verified.

### 6.1. Experimental Process

The temperature measurement experiment mainly consists of the following instruments and equipment: NiCr/NiSi thin-film thermocouple, K-type wire thermocouple, K-type compensation wire, DT9205A digital multimeter (ANRECSON, Shenzhen, China), automobile engine, stopwatch, etc. The schematic diagram of the experiment is shown in Figure 10. According to the performance of the automobile engine, the temperature can be measured in the range of about 20–120 °C.

The specific process of the temperature measurement experiment is as follows: Firstly, connect the NiCr/NiSi thin-film thermocouple and the K-wire thermocouple to the K-type compensation wire and the digital multimeter, respectively. Then, paste the NiCr/NiSi thin-film thermocouple and the K-wire thermocouple at the same adjacent position of the automobile engine, switch the digital multimeter to the voltage range, and record the voltage values of the two digital multimeters, respectively. Start the automobile engine, measure and record the value of the induced voltage every 10 s. When the value of the induced voltage does not change for a period of time, stop recording and complete a temperature measurement experiment. Repeat the above steps for 4–5 times, then measure and record the value of the induced voltage, respectively. During the measurement, try to increase the throttle frequently to change the rate of temperature change and record the voltage change.

After all the measurements are completed, according to the fitting curves of the static calibration experiments of the NiCr/NiSi thin-film thermocouple and the K-wire thermocouple, the fitting results of the NiCr/NiSi thin-film thermocouple are as follows:E = 0.03889T − 0.5264.(4)

The fitting results of K-wire thermocouple are as follows:E = 0.03923T − 0.7432,(5)

From this, the expression of calculated temperature can be obtained, and the NiCr/NiSi thin-film thermocouple isT = (E + 0.5264)/0.03889.(6)

The K-wire thermocouple isT = (E + 0.7432)/0.03923.(7)

According to the calculation formula of temperature, the voltage value recorded in the temperature measurement experiment is converted into the temperature value, and then drawn onto the data plot, with time as the abscissa and temperature as the ordinate.

### 6.2. Experimental Results

According to the above temperature measurement experiments of the NiCr/NiSi thin-film thermocouple and K-wire thermocouple, the temperature–time diagrams of the experiment can be obtained.

Figure 11 and Figure 12 show the results of the temperature measurement experiments. Figure 11 shows the experimental results of the K-wire thermocouple in five experiments with different colors. Figure 12 shows the experimental results of NiCr/NiSi thin-film thermocouple in five experiments.

During the first experiment, the engine was naturally warmed up without stepping on the accelerator of the car. When the automobile engine starts to work, the temperature rises gradually, and the induced thermoelectromotive force also rises gradually. With the continuous operation of the automobile engine, the overall temperature shows an upward trend, and the temperature–time curve rises. In the process of temperature measurement, when the value of the digital multimeter does not change for a period of time, that is, the induced thermoelectromotive force of the NiCr/NiSi thin-film thermocouple and the K-wire thermocouple does not change, it shows that the working temperature of the automobile engine is stable at a certain value. Then, the measurement is stopped, and the highest temperature of the stable operation of the engine is 94.9 °C. In the temperature–time diagram, the drawn curve finally tends to be horizontal, indicating that the measured temperature has reached a stable maximum.

In the second experiment, the automobile engine naturally warmed up without stepping on the accelerator. The temperature–time diagram obtained by the second measurement is basically the same as the result of the first experiment. The temperature increases steadily as a whole, and the slope of the curve is basically stable. When the temperature reaches above 80 °C, the slope of the curve fluctuates and changes slightly, but there is no obvious increase or decrease. Finally, the curve tends to be horizontal, indicating that the engine has reached a stable maximum working temperature of 107.9 °C.

In the third experiment, after the automobile engine works for a period of time, the accelerator of the automobile is continuously stepped on, which makes the temperature of the engine change. In the temperature–time diagram, before the temperature reaches 40 °C, the slope of the curve is small and basically unchanged. After the temperature reaches 40 °C, the temperature of the engine rises rapidly due to stepping on the accelerator, and the slope of the curve increases obviously, indicating that the temperature change rate increases. When the temperature reaches 80 °C, the engine works naturally again, and the slope of the curve decreases, which is almost the same as the initial slope, until the final curve tends to be horizontal, and the stable maximum working temperature of the automobile engine is 110.4 °C.

In the fourth experiment, after the automobile engine works for a period of time, the accelerator of the automobile is also continuously stepped on, so that the temperature of the engine changes, and the temperature change is measured again. In the temperature–time diagram, before the temperature reaches 35 °C, the curve changes smoothly, and the temperature change rate basically does not change. After the temperature reaches 35 °C, the slope of the curve increases obviously because the engine temperature change rate is increased by stepping on the accelerator. After the temperature reaches 80 °C, the slope of the curve decreases and tends to be flat again, which is roughly the same as the initial slope. Finally, the curve tends to be horizontal, and the stable maximum working temperature of the automobile engine is 107.9 °C.

In the fifth experiment, after the automobile engine worked for a period of time, we stepped on the accelerator of the automobile continuously, and then stepped on the accelerator of the automobile again after a period of time to measure the temperature change. In the temperature–time diagram, before the temperature reaches 38 °C, the automobile engine is in a natural working state, and the slope of the curve is small and basically unchanged. After the temperature reaches 38 °C, the temperature of the engine rises rapidly due to stepping on the accelerator, and the slope of the curve increases obviously, at which time the temperature change rate increases. When the temperature reached 80 °C, the slope of the curve increased slightly again due to stepping on the accelerator of the car again, and the temperature change rate increased. After a period of time, the engine was in a natural working state again, and the slope of the curve decreased to about the same as the initial slope. Finally, the curve tends to be horizontal, and the stable maximum working temperature of the automobile engine is 110.5 °C.

The temperature–time curves of the NiCr/NiSi thin-film thermocouple and the K-type wire thermocouple obtained by the above five temperature measurement experiments basically coincide, which can prove that the performance of the NiCr/NiSi thin-film thermocouple prepared in this paper is consistent with that of the K-type thermocouple, with relatively good accuracy and sensitivity. The measured maximum stable working temperature of the automobile engine is about 107.9 °C. When the measured temperature changes rapidly, the slope of the graph increases obviously, which shows that the NiCr/NiSi thin-film thermocouple has a good dynamic response to the temperature change.

According to the measurement results, the temperature–time curves of the NiCr/NiSi thin-film thermocouple and K-wire thermocouple are not completely coincident, and there is a certain error. It is found that with the increase in substrate temperature, the surface of the NiCr/NiSi thin-film thermocouple sample is flat, and the crystal particles are gradually dense. The granular structure on the film surface presents an obvious sharpening phenomenon (Huang et al., 2022 [22]; Liu, 2024 [23]; Lv, 2023 [24]), thus affecting the Zeebek coefficient of thermocouple, resulting in inaccurate temperature measurement and errors. When the temperature is measured by the thermocouple, the sensing part of the thermocouple will also carry out convection heat exchange with the surrounding fluid, radiation heat exchange with the environment, and heat conduction with the lead (Kou et al., 2023 [25]; Liu, 2018 [26]; Cui et al., 2017 [27]). Therefore, the thermocouple will be accompanied by convection error, heat conduction error, and radiation error in the process of measuring temperature. This means that the temperature displayed by the thermocouple cannot accurately represent the actual temperature of the measured object, and thus reduces the temperature measurement accuracy of the thermocouple.

## 7. Conclusions

According to the Seebeck effect, the NiCr/NiSi thin-film thermocouple was prepared by magnetron sputtering. In order to verify its performance and accuracy, the static calibration experiment and temperature measurement experiment of the NiCr/NiSi thin-film thermocouple and K-wire thermocouple were carried out.

In the static calibration experiment of the NiCr/NiSi thin-film thermocouple and K-type wire thermocouple, the temperature and thermal induced voltage data measured and recorded by the experiment are fitted by using the least square method, and the expression of K-type wire thermocouple is obtained as follows:E = 0.03923T − 0.7432.(8)

The expression of NiCr/NiSi thin-film thermocouple isE = 0.03889T − 0.5264,(9)
where T is the temperature measured by the thermocouple and E is the induced thermoelectromotive force of the thermocouple, the Seebeck coefficient of the K-type wire thermocouple is 39.10 μV/°C, the fitted correlation coefficient R^2^ is 0.9987, the Seebeck coefficient of the NiCr/NiSi thin-film thermocouple is 38.67 μV/°C, and the fitted correlation coefficient R^2^ is 0.9963, all of which have good linear relationships. According to the reference data, the Seebeck coefficient of a standard K-type thermocouple is 41.20 μV/°C. In the static calibration experiment, the errors between the Seebeck coefficient of the K-type wire thermocouple and NiCr/NiSi thin-film thermocouple and the Seebeck coefficient of standard K-type thermocouple are 1.97 μV/°C and 2.31 μV/°C, respectively, and the errors are within the acceptable range. The consistency and repeatability experiments were carried out by using a NiCr/NiSi thin-film thermocouple. The Seebeck coefficients of the NiCr/NiSi thin-film thermocouple were 39.10 μV/°C, 39.35 μV/°C, 38.84 μV/°C, 38.95 μV/°C and 39.22 μV/°C, respectively, with an average of 39.092 μV/°C, which was the same as the previous experiments. It can be proved that the prepared NiCr/NiSi thin-film thermocouple has good consistency and repeatability, and it also proves that the NiCr/NiSi thin-film thermocouple has good and stable performance.

In the temperature measurement of the NiCr/NiSi thin-film thermocouple and K-wire thermocouple, the temperature of the automobile engine was measured by using NiCr/NiSi thin-film thermocouple and K-wire thermocouple at the same time, and the dynamic response of the NiCr/NiSi thin-film thermocouple to temperature was verified. According to the expression of the static calibration experiment, the temperature expressions of the NiCr/NiSi thin-film thermocouple and K-wire thermocouple are obtained, respectively. The temperature expression of NiCr/NiSi thin-film thermocouple isT = (E + 0.5264)/0.03889.(10)

The temperature expression of K-wire thermocouple isT = (E + 0.7432)/0.03923.(11)

According to the measured thermal induced voltage values, the temperature is obtained by bringing them into expressions, respectively, and the temperature–time diagram is drawn. In the first two experiments, the automobile engine was made to work normally and naturally, and in the last three experiments, the temperature change rate was changed by stepping on the accelerator of the automobile. Among the five temperature–time curves drawn, the slope of the curve represents the rate of change in temperature. It can be seen that the change in temperature is relatively stable when the engine works normally and naturally, and the slope of the curve fluctuates in a small range but does not change obviously. When stepping on the automobile accelerator, the temperature change rate of the automobile engine increases, and the slope of the curve increases obviously. After stopping stepping on the automobile accelerator, the slope of the curve is equal to that at the beginning of natural work again. Finally, the curves tend to be horizontal, with a slope of 0, indicating that the engine has reached the highest temperature of stable operation at this time. After many measurements, the highest temperature of stable operation of anautomobile engine is about 107.9 °C. The temperature measurement experiments of NiCr/NiSi thin-film thermocouple and K-wire thermocouple can further prove that the NiCr/NiSi thin-film thermocouple prepared in this paper has good performance and high accuracy, and has good dynamic response to temperature changes.

## Figures and Tables

**Figure 1 micromachines-16-00249-f001:**
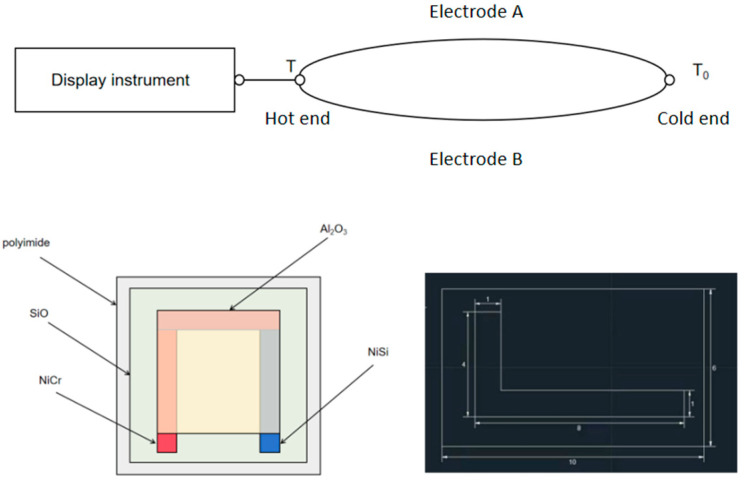
Schematic diagram of thermocouple working principle, structure diagram, and mask size diagram of NiCr/NiSi thin-film thermocouple.

**Figure 2 micromachines-16-00249-f002:**
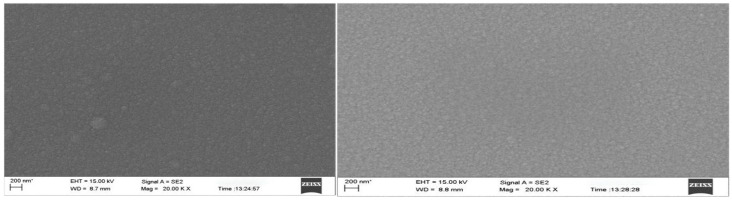
SEM observation of NiCr (**left**) and NiSi (**right**).

**Figure 3 micromachines-16-00249-f003:**
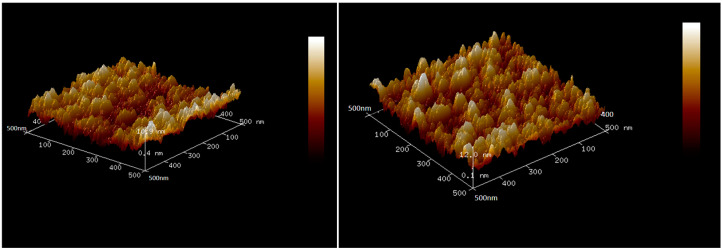
AFM observation of NiCr (**left**) and NiSi (**right**).

**Figure 4 micromachines-16-00249-f004:**
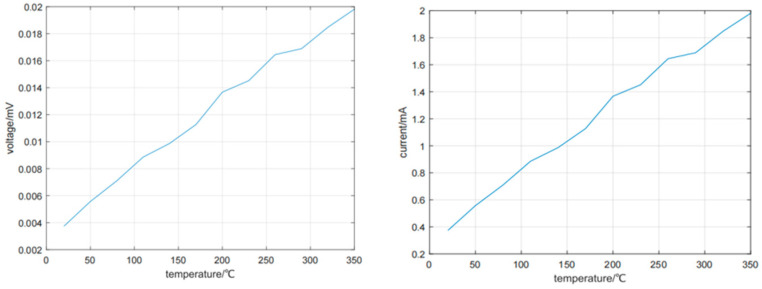
Test results of voltage (**left**) and current (**right**).

**Figure 5 micromachines-16-00249-f005:**
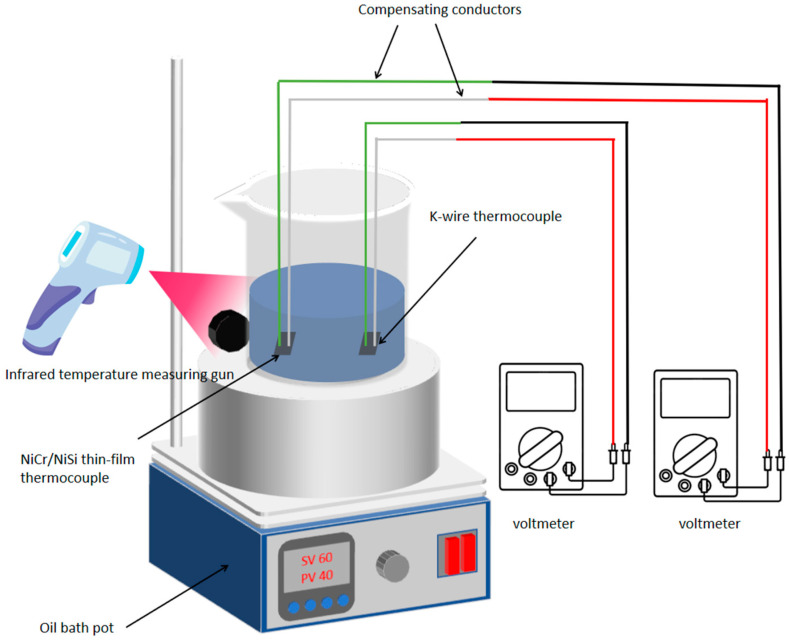
Schematic diagram of static calibration experiment of NiCr/NiSi thin-film thermocouple and K-wire thermocouple.

**Figure 6 micromachines-16-00249-f006:**
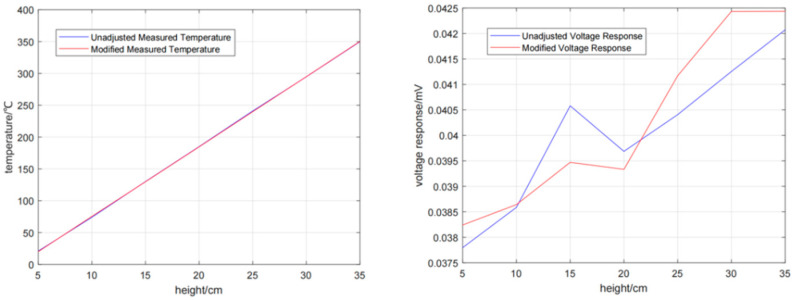
Temperature distribution curve and thermocouple output voltage.

**Figure 7 micromachines-16-00249-f007:**
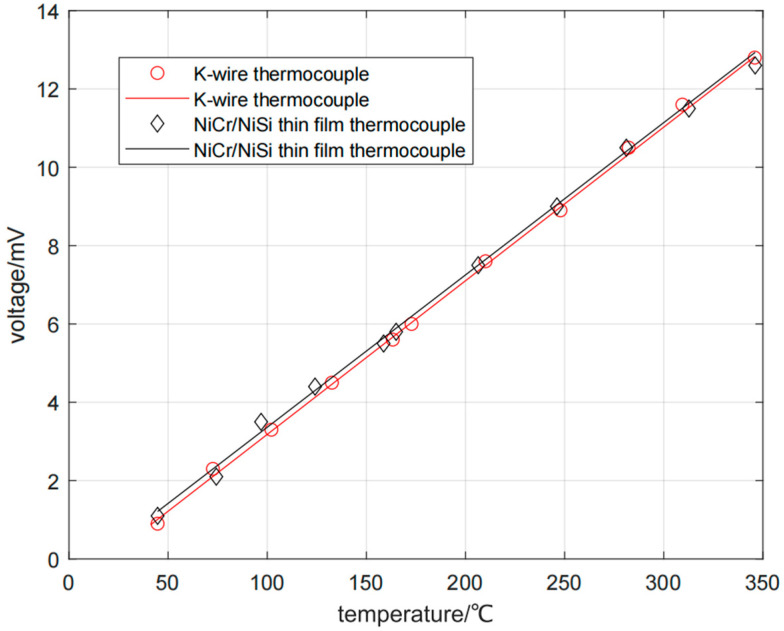
Fitting curves of K-type wire thermocouple and NiCr/NiSi thin-film thermocouple.

**Figure 8 micromachines-16-00249-f008:**
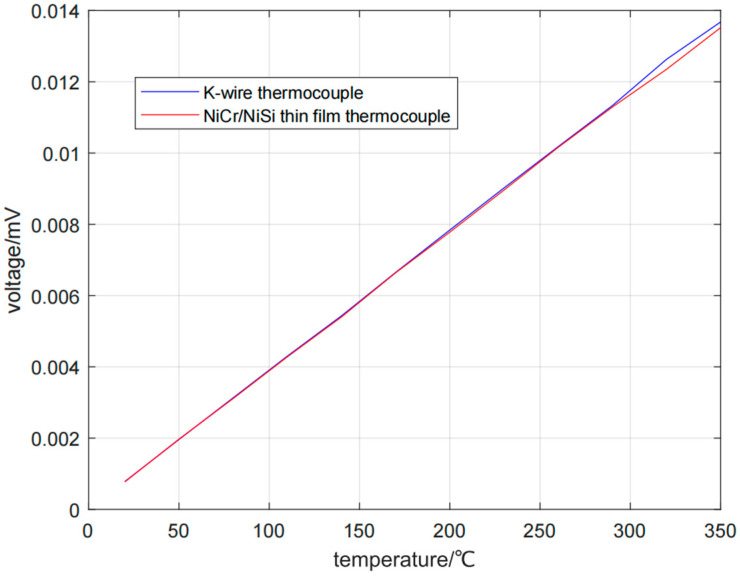
Temperature–voltage output curve.

**Figure 9 micromachines-16-00249-f009:**
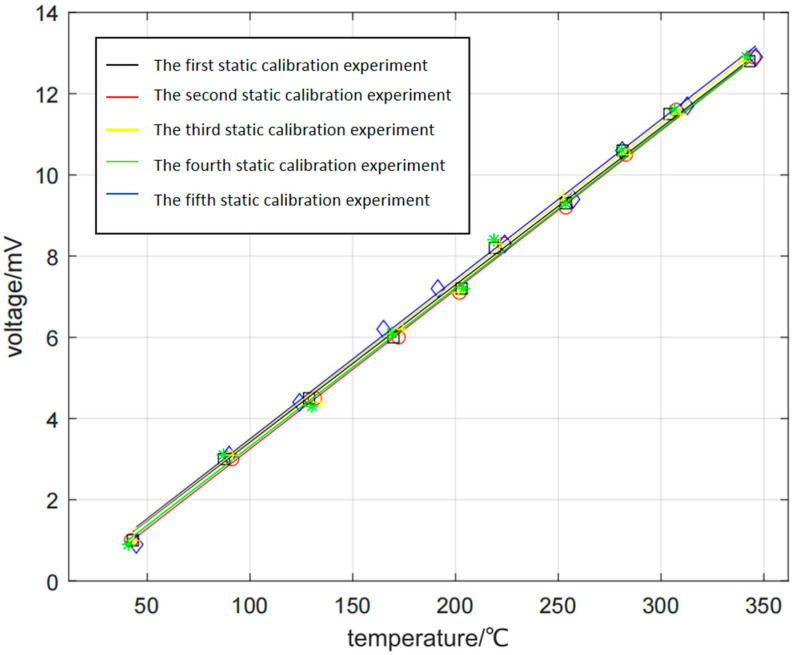
Fitting curve of NiCr/Nisi thin-film thermocouple.

**Figure 10 micromachines-16-00249-f010:**
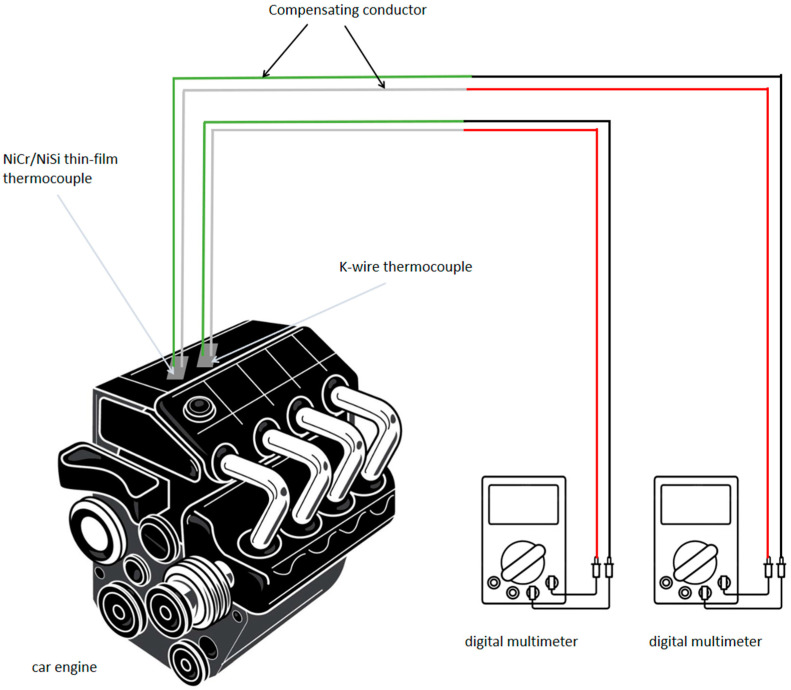
Schematic diagram of temperature measurement experiment of NiCr/NISI thin-film thermocouple and K-wire thermocouple.

**Figure 11 micromachines-16-00249-f011:**
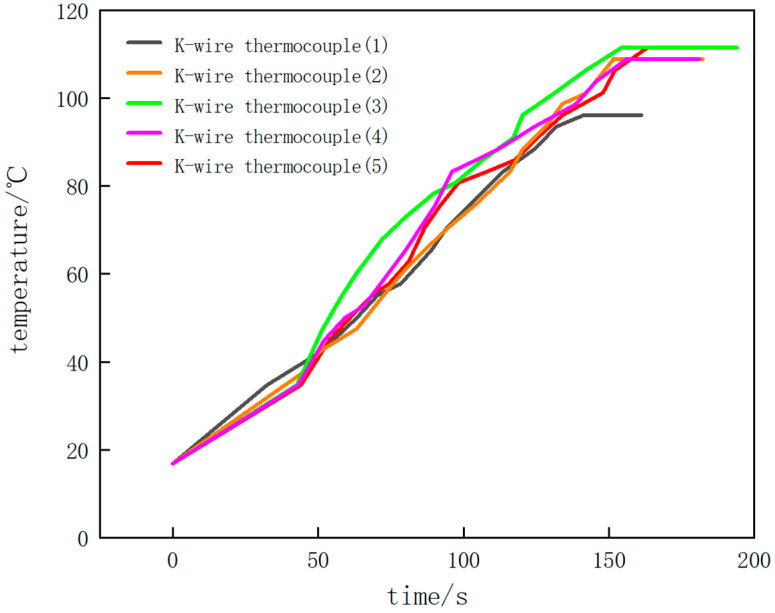
Temperature–time diagram of K-wire thermocouple of temperature measurement experiment.

**Figure 12 micromachines-16-00249-f012:**
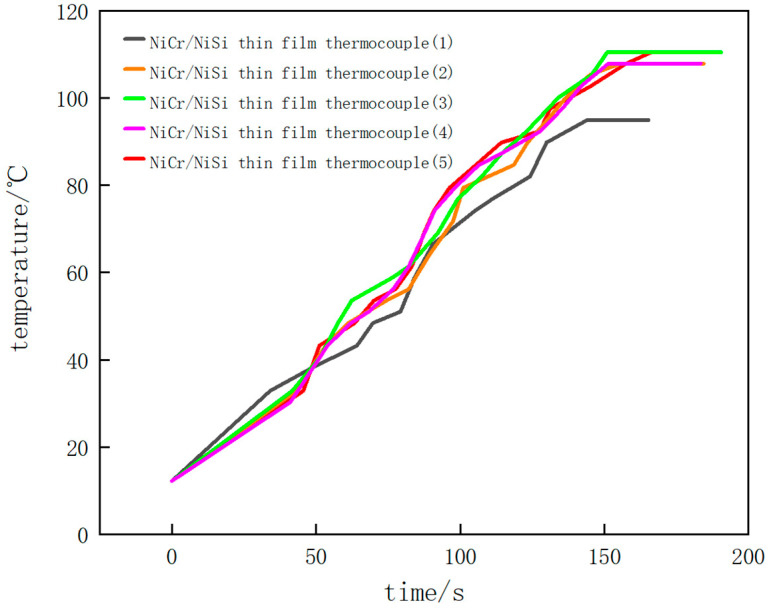
Temperature–time diagram of NiCr/NiSi thin-film thermocouple of temperature measurement experiment.

## Data Availability

The original contributions presented in the study are included in the article, further inquiries can be directed to the corresponding author.

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
