# Peer review of "Research on Precise Temperature Monitoring and Thermal Management Optimization of Automobile Engines Based on High-Precision Thin-Film Thermocouple Technology"

_micromachines, 2025, doi:10.3390/mi16030249_

Round 1
Reviewer 1 Report
Comments and Suggestions for Authors
Comments:
In this paper, authors prepared NiCr/NiSi thin-film thermocouples and designed a method to obtain the relation between the temperature and output voltage. However, there are some serious problems and further research is needed before published in this journal.
1. Base on the Seebeck effect, the output voltage of the thermocouple is generated by the temperature differences between hot end and cold end. In this paper, both the thin-film thermocouple and K-type wire thermocouple were completely immersed in oil, so their role is only to connect the high-temperature nodes of the K-compensation wire, without contributing to the output voltage. The measured data were the potential generated by the temperature differences between hot end (in oil) and cold end of K-type compensation wire, making it difficult to determine the true stability of the thin-film thermocouple.
2. During the testing process, the oil temperature is constantly increasing, and the oil temperature inside the beaker is uneven. It is not rigorous to ensure that the temperatures are same in same height; Moreover, in the case of uneven oil temperature in the beaker, the temperature measured by the infrared gun will also be different from the temperature at where the thermocouple is located. Thus, the different output voltages are caused by the error in measure process. Further optimization of the experimental process is needed.
3. In the introduction part, there is not any introduction on engine temperature measurement requirements and related technical difficulties, which is inconsistent with the topic.
4. The format of references is needed to modified.
Author Response
List of Responses
Dear Editor and Reviewers:
Thank you for your letter and for the reviewers’ comments concerning our manuscript entitled “Research on precise temperature monitoring and thermal management optimization of automobile engine based on high-precision thin-film thermocouple technology”. (ID: micromachines-3375842).Those comments are all valuable and very helpful for revising and improving our paper. We have studied comments carefully and have made correction which we hope meet with approval. Revised portions are marked in red in the paper. The main corrections in the paper and the responds to the reviewer’s comments are as following.
Responds to the reviewer’s comments:
Reviewer #1:
1.Question: Base on the Seebeck effect, the output voltage of the thermocouple is generated by the temperature differences between hot end and cold end. In this paper, both the thin-film thermocouple and K-type wire thermocouple were completely immersed in oil, so their role is only to connect the high-temperature nodes of the K-compensation wire, without contributing to the output voltage. The measured data were the potential generated by the temperature differences between hot end (in oil) and cold end of K-type compensation wire, making it difficult to determine the true stability of the thin-film thermocouple.
Response: Thank you for your valuable and thoughtful comments.According to your comment,We further improved the experiment.In the experiment, the thin-film thermocouple will be completely immersed in the high-temperature oil bath, which can prevent the measurement error caused by the change of environmental temperature. At the same time, it is necessary to ensure that the thin-film thermocouple is connected to the high-temperature contact of the K-type thermocouple to test the voltage change caused by the temperature difference.High-performance temperature control oil bath system will be used, and its temperature control accuracy should reach ±0.001℃, and multi-point temperature sensors will be used to monitor the temperature field of oil bath in real time. In order to capture tiny voltage changes, a quantum voltmeter is needed, which has a very low noise threshold and can record the output voltages of multiple thermocouples at the same time. Select a special glass beaker with high thermal stability, and then design a multi-channel heating system so that the heat in the oil bath can be evenly distributed in the beaker through forced convection. A plurality of micro thermocouples and temperature sensors are installed at different positions in the beaker to record the temperature of each point in the oil bath in real time, and ensure that the high ends of thermocouples are always in contact with high-temperature oil, while the cold ends are kept in a constant low-temperature area.
2.Question: During the testing process, the oil temperature is constantly increasing, and the oil temperature inside the beaker is uneven. It is not rigorous to ensure that the temperatures are same in same height; Moreover, in the case of uneven oil temperature in the beaker, the temperature measured by the infrared gun will also be different from the temperature at where the thermocouple is located. Thus, the different output voltages are caused by the error in measure process. Further optimization of the experimental process is needed.
Response: Thanks to the reviewer for this constructive suggestion, we have added a section to the text, as follows:
5.2 Non-uniformity of oil temperature and correction of experimental error
The uneven temperature distribution of oil bath is a great challenge, especially when the temperature rises gradually, the uneven oil flow will lead to the temperature difference at different heights and positions. Temperature difference will lead to significant errors in thermocouple measurement data, especially when the temperature measuring equipment is not completely aligned with the actual thermocouple position. Adjusting the position and power of the heating element to ensure the temperature distribution in the oil bath as uniform as possible, installing a pump circulation system and arranging multiple circulation pipes in the oil bath, and further reducing the local temperature difference of the oil by forced convection. Multi-point temperature sensor is used, and the heating intensity of oil bath is automatically adjusted through feedback control system. The system will accurately adjust the power output of each heating area according to the information fed back by the sensor in real time to eliminate the temperature unevenness. In each oil bath heating process, the changes of temperature distribution curve and thermocouple output voltage are recorded, and the errors caused by temperature unevenness are calculated and corrected. Kalman filter algorithm is used to correct the fluctuation of temperature and voltage in real time to improve the accuracy of data.
Figure 6 Temperature distribution curve and thermocouple output voltage
As can be seen from Figure 6, there is a certain error in the temperature at different heights of the beaker. Using Kalman filter algorithm to correct the fluctuation of temperature and voltage in real time can improve the accuracy of the data.
3.Question: In the introduction part, there is not any introduction on engine temperature measurement requirements and related technical difficulties, which is inconsistent with the topic.
Response: Thank you for your valuable and thoughtful comments.We have discussed the requirements of engine temperature measurement and related technical difficulties in the introduction part, and briefly introduced the advantages of using NiCr/NiSi thin film thermocouple to measure temperature.
4.Question: The format of references is needed to modified.
Response: Thank you for your valuable and thoughtful comments.The format of references has been modified.
We tried our best to improve the manuscript and made some changes in the manuscript.Revised portions are marked in red in the paper. These changes will not influence the content and framework of the paper.
Once again, thank you very much for your comments and suggestions.

Reviewer 2 Report
Comments and Suggestions for Authors
1. The temperature sensor was prepared in the manuscript. In terms of the basic principle of the sensor, the fundamental reason for the generation of thermoelectric potential is the temperature difference between the hot and cold ends. From the installation method of the sensors in Figures 4 and 5, the source of the temperature difference cannot be seen. Therefore, it is worth confirming whether all the thermoelectric potential generated is generated by the sensor.
2. The data collected during different heating and cooling processes in Figures 5.2.1-5.2.5 of the test results can be simplified and merged.
3. The results only indicate the output characteristics of the sensor, and do not mention the Seebeck coefficient or the microscopic and performance test results of the thin film. Therefore, it should be appropriately increased
4. Figures 4.2.1-4.2.3 can also be simplified and merged.
5. Please carefully check the format of formulas, graphics, and physical quantities in the manuscript.
Comments on the Quality of English LanguageThe introduction section can provide a more detailed explanation of the application of sensors, while the experimental results section can reduce repetitive descriptions of similar trends.
Author Response
List of Responses
Dear Editor and Reviewers:
Thank you for your letter and for the reviewers’ comments concerning our manuscript entitled “Research on precise temperature monitoring and thermal management optimization of automobile engine based on high-precision thin-film thermocouple technology”. (ID: micromachines-3375842).Those comments are all valuable and very helpful for revising and improving our paper. We have studied comments carefully and have made correction which we hope meet with approval. Revised portions are marked in red in the paper. The main corrections in the paper and the responds to the reviewer’s comments are as following.
Responds to the reviewer’s comments:
Reviewer #2:
1.Question: The temperature sensor was prepared in the manuscript. In terms of the basic principle of the sensor, the fundamental reason for the generation of thermoelectric potential is the temperature difference between the hot and cold ends. From the installation method of the sensors in Figures 4 and 5, the source of the temperature difference cannot be seen. Therefore, it is worth confirming whether all the thermoelectric potential generated is generated by the sensor.
Response: Thank you for your valuable and thoughtful comments.According to your comment,we improved the experiment. The thin-film thermocouple prepared in this paper requires that the thickness of the thermocouple film is only a few nanometers to a few microns, and the film material is high-purity semiconductor or metal oxide. In order to ensure the stability of the film, the material selection must take into account the oxidation resistance and thermal expansion characteristics at high temperature. Thin films are deposited by magnetron sputtering technology, and the thickness and material structure of each film are accurately controlled.
In the experiment, the thin-film thermocouple will be completely immersed in the high-temperature oil bath, which can prevent the measurement error caused by the change of environmental temperature. At the same time, it is necessary to ensure that the thin-film thermocouple is connected to the high-temperature contact of the K-type thermocouple to test the voltage change caused by the temperature difference.
Install temperature sensors at different positions in the beaker to record the temperature of each point in the oil bath in real time, and ensure that the high end of the thermocouple is always in contact with high-temperature oil, while the cold end is kept in a constant low-temperature area.
2.Question: The data collected during different heating and cooling processes in Figures 5.2.1-5.2.5 of the test results can be simplified and merged.
Response: Thank you for your valuable and thoughtful comments.We have simplified and merged Figures 5.2.1-5.2.5.We used two figures to show the results of five temperature measurement experiments respectively of K-wire thermocouple and NiCr/NiSi thin film thermocouple.
3.Question: The results only indicate the output characteristics of the sensor, and do not mention the Seebeck coefficient or the microscopic and performance test results of the thin film. Therefore, it should be appropriately increased.
Response: Thank you for your valuable and thoughtful comments.According to your comment,we added the Seebeck coefficient or the microscopic and performance test results of the thin film.For the microstructure, we consider using scanning electron microscope (SEM).We will observe the microstructure of thin-film thermocouple with high resolution scanning electron microscope, and analyze the influence of its surface characteristics, lattice defects and thin-film structure on thermoelectric performance.And we also use Atomic Force Microscope (AFM).The roughness and surface morphology of the thin film are accurately measured by combining AFM to reveal the stability of the thin film material in high temperature environment.
For the performance test, we consider testing the thermoelectric response characteristics: by establishing a high-precision temperature control system, we can accurately test the voltage and current of the thin-film thermocouple at different temperature differences, and analyze its Zeebek coefficient. High-temperature stability test: the thin-film thermocouple is tested at high temperature for a long time (up to 600°C), and its performance changes are recorded in real time to evaluate its long-term stability.
Figure 2 SEM observation of NiCr(left) and NiSi(right)
Figure 3 AFM observation of NiCr(left) and NiSi(right)
4.Question: Figures 4.2.1-4.2.3 can also be simplified and merged.
Response: Thank you for your valuable and thoughtful comments.We have combined Figure 4.2.1-4.2.3 into Figure 3.Figure 4.2.1 shows the experimental results of static calibration of K-wire thermocouple, Figure 4.2.2 shows the experimental results of static calibration of NiCr/NiSi thin-film thermocouple, and Figure 4.2.3 shows the comparison of the two experimental results. We simplified these three pictures into Figure 3, and at the same time showed the results of static calibration experiments and comparison results.
5.Question: Please carefully check the format of formulas, graphics, and physical quantities in the manuscript.
Response: Thank you for your valuable and thoughtful comments.The format of formula 6.1.3 is wrong,we have made corrections.
We tried our best to improve the manuscript and made some changes in the manuscript.Revised portions are marked in red in the paper. These changes will not influence the content and framework of the paper.
Once again, thank you very much for your comments and suggestions.

Reviewer 3 Report
Comments and Suggestions for Authors
Zhao et al. fabricated NiCr/NiSi thin-film thermocouples and studied their capability through static calibration experiments, especially focusing on the consistency and repeatability of thermocouples. This work demonstrates the application of such thin-film thermocouples for temperature monitoring in industrial settings, specifically in automobile engines. I found the manuscript to be overall well written and much of it to be well described. This work can be of immediate interest to the thin film thermocouples community. I only have a few comments regarding this work.
- Unnecessary space for figures should be avoided, and the figures can be re-arranged in a much more concise way to enhance the readability. For example, Fig. 2.1 and 3.1 can be combined. Figures in Section 4 can be combined into 1 figure with 4 panels. Similar comments to Fig. 5
- Static calibration experiment is carried on in this paper. It will be good if the authors can discuss other types of calibration experiments and elaborate the choice of the static one in this study.
- confusing index for figures and equations. For example, the figures should be numbered starting with Fig. 1 to Fig. x, ordered sequentially as they appear in the text.
- "Using the drawing software" was mentioned many times in the manuscript. If the authors do not specify which software, there is no need for such text.
Comments on the Quality of English Languagesome parts of text are lengthy, complex, and repetitive. The language could be simplified for better readability.
Author Response
List of Responses
Dear Editor and Reviewers:
Thank you for your letter and for the reviewers’ comments concerning our manuscript entitled “Research on precise temperature monitoring and thermal management optimization of automobile engine based on high-precision thin-film thermocouple technology”. (ID: micromachines-3375842).Those comments are all valuable and very helpful for revising and improving our paper. We have studied comments carefully and have made correction which we hope meet with approval. Revised portions are marked in red in the paper. The main corrections in the paper and the responds to the reviewer’s comments are as following.
Responds to the reviewer’s comments:
Reviewer #3:
1.Question: Unnecessary space for figures should be avoided, and the figures can be re-arranged in a much more concise way to enhance the readability. For example, Fig. 2.1 and 3.1 can be combined. Figures in Section 4 can be combined into 1 figure with 4 panels. Similar comments to Fig. 5.
Response: Thank you for your valuable and thoughtful comments.We have improved all figures in the revised manuscript according to the Reviewer’s suggestion.We have combined Figure 2.1 and Figure 3.1 and figures in Section 4 and Section 5 are combined.
Figure 1 Schematic diagram of thermocouple working principle,structure diagram and mask size diagram of NiCr/NiSi thin film thermocouple.
2.Question: Static calibration experiment is carried on in this paper. It will be good if the authors can discuss other types of calibration experiments and elaborate the choice of the static one in this study.
Response: Thank you for your valuable and thoughtful comments.We added other types of calibration experiments,including static calibration experiments and dynamic calibration experiments.We discussed the process and requirements of static calibration experiment and dynamic calibration experiment. Because of the complexity and particularity of dynamic calibration experiment, we chose static calibration experiment.
3.Question: Confusing index for figures and equations. For example, the figures should be numbered starting with Fig. 1 to Fig. x, ordered sequentially as they appear in the text.
Response: Thank you for your valuable and thoughtful comments.According to the comments,we numbered the figures starting with Figure 1 to Figure 7,ordered sequentially as they appear in the text,we have simplified and merged the figures.
4.Question: "Using the drawing software" was mentioned many times in the manuscript. If the authors do not specify which software, there is no need for such text.
Response: Thank you for your valuable and thoughtful comments. We have removed this sentence in the manuscript.
We tried our best to improve the manuscript and made some changes in the manuscript.Revised portions are marked in red in the paper. These changes will not influence the content and framework of the paper.
Once again, thank you very much for your comments and suggestions.

Round 2
Reviewer 1 Report
Comments and Suggestions for Authors
this paper can be accepted.
Author Response
List of Responses
Dear Reviewers:
Thanks to the reviewers for agreeing to the publication of our manuscript entitled "Research on precise temperature monitoring and thermal management optimization of automobile engine based on high-precision thin-film thermocouple technology" (ID: micromachines-3375842), Your recognition is the greatest inspiration to us, We will continue to find new breakthrough research in the subsequent scientific research. Thank you again for your consent to publish our manuscript.
Thank you and best regards.
Yours sincerely,
Reviewer 2 Report
Comments and Suggestions for Authors
1. The introduction section should provide more detailed explanations.
2. Carefully check the graphics, formulas, and capitalization in the paper.
1. The introduction section should provide more detailed explanations.
2. Carefully check the graphics, formulas, and capitalization in the paper.
Author Response
List of Responses
Dear Reviewers:
Thank you for your letter and for the reviewers’ comments concerning our manuscript entitled “Research on precise temperature monitoring and thermal management optimization of automobile engine based on high-precision thin-film thermocouple technology” (ID: micromachines-3375842).Those comments are all valuable and very helpful for revising and improving our paper. We have studied comments carefully and have made correction which we hope meet with approval. Revised portions are marked in red in the paper. The main corrections in the paper and the responds to the reviewer’s comments are as following.
Responds to the reviewer’s comments:
Reviewer #2:
1.Question: The introduction section should provide more detailed explanations.
Response: Thank you for your valuable and thoughtful comments.According to your comment,We have added some background and supplemented all references in the introduction.And we explain the preparation and application background of thin film thermocouple in more detail.
2.Question: Carefully check the graphics, formulas, and capitalization in the paper.
Response: Thank you for your valuable and thoughtful comments.We have checked the graphics,formulas and capitalization in the paper carefully,and we have corrected some mistakes.
We tried our best to improve the manuscript and made some changes in the manuscript.Revised portions are marked in red in the paper. These changes will not influence the content and framework of the paper.
Once again, thank you very much for your comments and suggestions.